# DynCNN: An Effective Dynamic Architecture on Convolutional Neural Network for Surveillance Videos

## Abstract

The large-scale surveillance video analysis becomes important as the development of intelligent city. The heavy computation resources neccessary for state-of-the-art deep learning model makes the real-time processing hard to be implemented. This paper exploits the characteristic of high scene similarity generally existing in surveillance videos and proposes dynamic convolution reusing the previous feature map to reduce the computation amount. We tested the proposed method on 45 surveillance videos with various scenes. The experimental results show that dynamic convolution can reduce up to 75.7% of FLOPs while preserving the precision within 0.7% mAP. Furthermore, the dynamic convolution can enhance the processing time up to 2.2 times.

## 1 Introduction

Nowadays, with the development of deep learning technologies, large-scale surveillance video analysis for intelligent city draws more and more attention in the real world applications, for instance, in person re-identification (Ahmed et al. (2015)), vehicle re-identification (Liu et al. (2016))(Shen et al. (2017)), pedestrian detection (Tian et al. (2014)) and crowd segmentation (Kang & Wang (2014)), etc. However, these deep learning methods are extremely computationally expensive—state-of-the-art methods for object detection performed on a state-of-the-art NVIDIA P100 GPU run at 10-80 frames per second. Although processing on one video can be done by utilizing one to several top-performing hardware equipments, it still a challenge for large-scale actual deployment—it would cost over 5 billion USD in hardware to analyze over 4 million CCTVs in real time in the UK alone when considering these computational overheads in context (News (2015)).

Those real world applications for intelligent city employed the state-of-the-art network architectures from the ILSVRC competition, e.g. VGG (Simonyan & Zisserman (2014)), GoogLeNet (Szegedy et al. (2014)), and ResNet (He et al. (2015)) as their feature extraction architecture. However, these powerful networks tend to be resource-hungry models with high computational demands at inference time to win the competition—processing an one-minute surveillance video through VGG16 network costs 23 TFLOPs. To make the feature extraction architectures more effective in terms of surveillance video, this paper aims to explore ways to reduce the calculation on feature extraction.

Figure 1 shows two frame clips of a surveillance video with its corresponding feature map through $3\times3$ kernel. It is observed that most of the scene between two adjacent frames in the surveillance video are almost the same with merely about 2% to 15% difference. Due to the linear characteristic of convolution, the derived feature maps preserve the similarity of input video frames—it can reduce significant amount of calculation if the previous feature map can be reused for the current feature map. Suppose that the frame size is $512\times512$ while the amount of the changing pixels in feature map is 52k, the calculation can be reduced from 7 MFLOPs to 1.4 MFLOPs, that is 5 times saving ratio.

In this paper, we exploit the key characteristic of surveillance video—high scene similarity—and propose a new architecture *dynamic convolution* which can be directly applied to off-the-shelf model without retraining the weights. Dynamic convolution can inspect the inter-frame variation and thus predict the changing pixels in each corresponding feature map. Accordingly, convolution performs only on those predicted parts to achieve less computation while not affecting the accuracy. Finally,

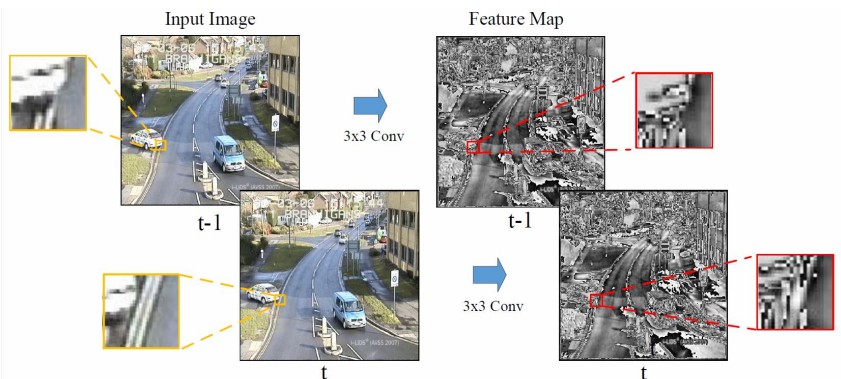

Figure 1: The subtle difference in the feature map.

we apply dynamic convolution to one of the state-of-the-art object detector Single Shot MultiBox Detection (Liu et al. (2015)) and test on several surveillance video datasets including overall 45 videos. In addition, we conduct a further analysis in the FLOP computation and execution time of each convolutional layer to improve the understanding of dynamic convolution in practice.

## 2 PRIOR WORKS

For the last several years, tuning deep learning neural architectures to strike an optimal balance between precision and processing time has been an area of active research. Many methods have been proposed to reduce storage space and computing resources and also made great progress from theoretical research to platform implementation. Basically, they can be divided into three major techniques, effective model design, pruning, and quantization.

**Effective model design** MobileNets (Howard et al. (2017)) is a lightweight network architecture proposed by Google for mobile deployment. The main idea is to disassemble the original convolution calculation into two parts: depthwise convolutions and pointwise convolutions. Although the calculation amount on convolution can greatly be reduced, the precision would be decreased. In addition, the feature map accordingly increases. The increasing data transfer followed by feature map makes MobileNets hard to perform on GPU. To improve the precision, ShuffleNet (Zhang et al. (2017)) proposed two new strategies channel shuffle and pointwise group convolutions based on the concept that helps the information flowing across feature channels. On the other hand, Inception (Szegedy et al. (2016)) introduces bottleneck structure to approximate sparse structures into several dense sub-matrices to achieve more efficient use of computing resources. In the ultra-lightweight neural network SqueezeNet (Iandola et al. (2016)), the 1x1 and 3x3 convolution kernels are extensively used compared with Inception to achieve more compression on calculation amount.

**Pruning** Besides of the model design, pruning is another efficient way to reduce storage space and computing resources by removing redundant parts of model. Li et al. (2016) proposed a method for cropping filters of convolutional layer. This method determines the importance of filter by the magnitude of the absolute sum of weights on the filter. After that, the model needs to be trained again. Without lossing greatly precision, such method can effectively reduce the complexity of the model. However, heavy analysis on each convolution layer costs lots of time and may occur the concern of one-case model. Liu et al. (2017) used another factor, the scaling coefficient $\gamma$ in batch normalization, to determine the importance of the filter.

**Quantization** Network weight sharing quantization is also an important type of network compression method which focuses on the weight representation. The nature of this technique is to find the center value which can represent the original weight distribution by clustering method. Deep Compression proposed by Han et al. (2015) clusters the weights and replaces them by the center value of each cluster. The transformation between the weights and center values is stored in a codebook. Such transformation greatly compresses the weight representation from a 32-bit floating point number to a short bit number.

## 3 DYNAMIC CONVOLUTION

Dynamic convolution is an architecture which applies to each layer of convolutional neural network. Figure 2 illustrates a convolution neural network with our dynamic convolution, called dynamic convolutional neural network (DynCNN). The work flow of dynamic convolution consists of three main parts: 1) frame differencing; 2) prediction; 3) dyn-convolution. Let $IDM_t$ denote the input difference map of $t$-th frame ($Frame_t$) derived by frame differencing and $iDM_t$ denote the inner difference map derived from the previous inner difference map or input difference map by prediction. Dyn-convolution preserves the feature map (FM) from the previous layer and selectively do convolution according to the position which the inner difference map indicates in order to update the feature map.

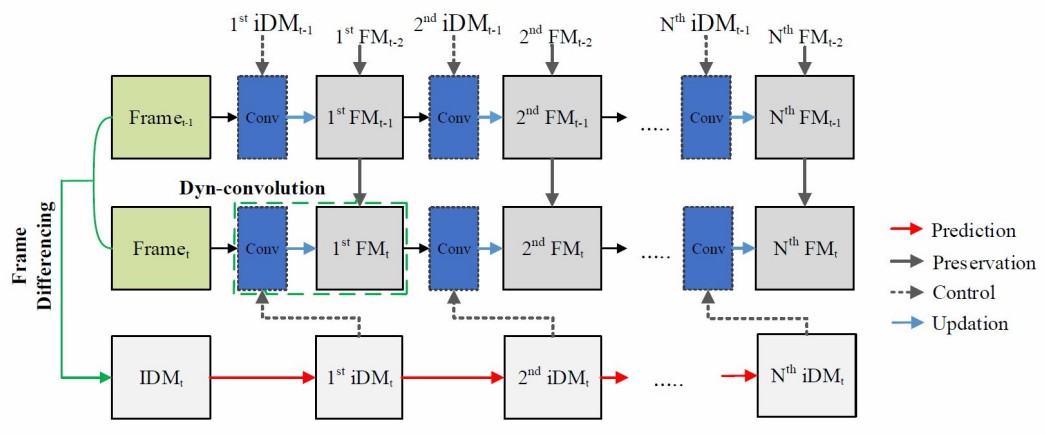

Figure 2: DynCNN Architecture

### 3.1 FRAME DIFFERENCING

The frame differencing method (Liu & Hou (2012)) is often used in the moving object detection and segmentation methods. In this work, we employ it to inspect the inter-frame variation between two adjacent frames. The basic concept of frame differencing is to subtract two frames to calculate the difference of each pixel. However, in most surveillance video sequences there exist speckle noise which severely affects the difference value. Accordingly, the difference map, as referred to input difference map (IDM), is derived by thresholding the result of frame differencing with the following statement:

$$D(i,j) = \begin{cases} 1, & |\Delta I(i,j)| < \Theta_{IDM} \\ 0, & otherwise \end{cases} \tag{1}$$

where $\Theta_{IDM}$ denotes the threshold of the statement. The pixel at $(i,j)$ is denoted as a changed pixel when $D$ is determined as 1 and we called it *changing point*. $\Delta I(i,j)$ in (1) represents the result of frame differencing as the following expression:

$$\Delta I(i,j) = I_{Curr}(i,j) - I_{Prev}(i,j) \tag{2}$$

where $I_{Curr}(i,j)$ and $I_{Prev}(i,j)$ in (2) denote the pixel intensity at position $(i, j)$ of the current and previous frame respectively.

The value of the threshold $\Theta_{IDM}$ is an important factor which determines the amount of convolution calculation. Therefore, the effect of the threshold on the speed and accuracy during the inference phase will be well analyzed in Section 4.

### 3.2 PREDICTION

In the convolution process, there exists a diffusion effect in the result if the input is changed. Suppose that the kernel size of convolution is $3 \times 3$, each pixel of the result is determined by the corresponding

9 pixels from the input—the pixel at the same position and its 8 peripheral pixels. On the contrary, when a certain pixel in the input is changed, 9 pixels in the output which the pixel involves will be affected simultaneously and are denoted as *impacted pixels*. Based on the characteristic of diffusion effect in convolution process, the position of impacted pixels in each feature map can be predicted by using dilation operation and recorded on the inner difference map.

### 3.3 DYN-CONVOLUTION

The iDM records the position of the impacted pixel of the FM and also implicates which pixel value of the FM needs to be updated. To update the value, the pixels of the previous FM which contributes to it will be re-convoluted and are denoted as *needed pixels* while this process is called *dyn-convolution*. In this work, we employ the lastest cuDNN library developed by Nvidia to optimize the performance on calculation speed. However, the convolution calculation provided by the library is only allowed for contiguous blocks instead of discrete blocks with given position. Therefore, we create another memory space, denoted as *new array*, in GPU to store those needed pixels for the requirement of library.

### 3.4 PRACTICAL CONCERN AND IMPLEMENTATION

In real surveillance video application, although the calculation reduction on convolution is the main concern of speeding up the overall processing time, the data transfer is another important factor which contributes to the time. Dyn-convolution is intuitive but rude and also not efficient in GPU. Accordingly, we apply two strategies to improve the overall processing time, *cell-based partition* and *continuous convolution*.

**Cell-based Partition** Compared with pixel-based map iDM, each FM is divided into several block cells. If any impacted pixel exists in the cell, the cell will be noted as "impacted" and recorded on *cell difference map* (CDM). The dyn-convolution thus performs according to the cell difference map rather than the pixel difference map—the new array stores the needed pixels of impacted cells. Figure 3 illustrates the difference between the naive method and the improved method.

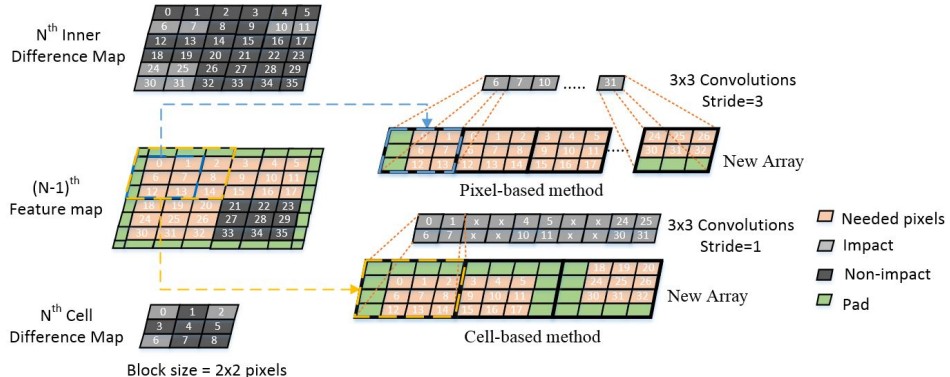

Figure 3: The diagram of data transfer. In the pixel-based method, the amount of needed pixels for one cell (e.g. $2 \times 2$ pixels) is $4 \times 9$ while the cell-based method requires $4 \times 4$ needed pixels, which greatly reduces the amount of data transfer.

In this strategy, the data in new array may contains "non-needed" pixels when the cell involves non-impacted pixels. In addition, new array performs unnecessary convolution on the boundary of two needed pixel blocks, where generates don't-care feature value x. Nevertheless, the needed pixels are reused to reduce the amount of data transfer and the characteristic of cuDNN library—convolution costs less processing time when stride is 1—is contemplated to speed up the convolution process, which indicates that less sacrification on computation while greatly improving the processing time.

**Continuous Convolution** Another part which costs processing time is the layer number. It multiplies the data transfer on a new array. We partition whole convolution layer by pooling process and

group the convolution layers up, called *convolution group*. For each convolution group, the convolution structure is modified from "once padding before once convolution" to "continuous padding first then continuous convolution". Continuous convolution aims at directly updating the last layer of the convolution group rather than updating each layer to further reduce the amount of data transfer on new array. Although the updated value may be blemished by the don't-care value x in continuous convolution, it causes insignificant effect on the overall precision while significantly saving the processing time.

## 4 EXPERIMENTS

This section will evaluate the performance of the dynamic convolution by employing the Single Shot MultiBox Detector $512 \times 512$ model [1] (SSD512) on GTX1080 GPU with cuDNN v5.1.

**Ground Truth.** The existing benchmarks with ground truth for multi-object detection are designed to evaluate the performance of CNN model on still image evaluation, such as Microsoft COCO (Lin et al. (2014)) and PASCAL VOC datasets (Everingham et al. (2010)). However, such benchmarks tested on the surveillance video are not given the ground truth. To well evaluate our proposed dynamic convolution, the ground truth is defined as the detection result of SSD512 model on surveillance video. We choose *VOC712Plus_SSD_512x512_iter_240000*—trained on VOC2007, VOC2012 and COCO, 240000 step iterations, 83.2% mAP on *VOC2007 test*—as the weight parameters.

**Baseline.** Dynamic convolution is a mechanism aimed at reducing the computation of original CNN model on surveillance video without affecting the performance. In the experiments, we choose the weight parameters *VOC0712_SSD_512x512_iter_120000*—trained on VOC2007 and VOC2012, 120000 step iterations, 79.8% mAP on *VOC2007 test*—as our baseline to evaluate our proposed method. The performance critiria is *how close the precision is compared to the baseline*.

**Dataset.** The dynamic convolution is tested on four surveillance video datasets: PETS2009 (Ferryman & Ellis (2014)), AVSS2007 (AVS (2007)), VIRAT (Oh et al. (2011)) and some videos from Youtube. In the surveillance video, person and car are the main objects which appear in the scene and are the most significant targets for surveillance application. Therefore, the performance in the experiments focuses on the objects of person and car. Table 1 lists the charascteristics of four surveillance video datasets, where the scene similarity ratio is defined as the ratio of the number of non-changing points to frame size.

Table 1: Comparison of characteristics of datasets

|  | PEST2009 | AVSS2007 | VIRAT | Youtube |
|---|---|---|---|---|
| Resolution | 768x576 | 720x576 | 1920x1080 | 1280x720 |
| Avg. Person Height in Pixels | 75 | 193 | 139 | 149 |
| Avg. Person to video height ratio | 13% | 34% | 13% | 21% |
| Avg. Car Height in Pixels | 49 | 66 | 104 | 98 |
| Avg. Car to video height ratio | 20% | 6% | 10% | 14% |
| Avg. Scene Similarity Ratio ($\Theta_{IDM} = 20$) | 96.6% | 97.4% | 99.9% | 97.4% |

### 4.1 ON PETS 2009

The PETS 2009 dataset is a benchmark which focuses on several challenges for crowd analysis in public area including the estimation of crowd person count and density, tracking of individual(s) within a crowd, and detection of flow and crowd events. In this dataset, we choose 20 difference scene videos totally involving 4,884 photos as test data. Table 2 presents the result of the performance on objects person and car with the threshold of frame differencing setting at 20. It is clear to see that the proposed dynamic convolution can preserve the precision with respect to baseline, where the difference is only 0.06% and 0.21% on person and car respectively.

---

[1]available at `https://github.com/weiliu89/caffe/tree/ssd`

Table 2: Average precision results on PETS2009

| Class-Person | TotalGT | TotalPred | TruePositives | FalsePositives | **AvgPrecision** |
|---|---|---|---|---|---|
| Baseline | 40720 | 39357 | 35202 | 4155 | **80.54%** |
| DynCNN | 40720 | 39227 | 35047 | 4180 | **80.48%** |
| Class-Car | TotalGT | TotalPred | TruePositives | FalsePositives | **AvgPrecision** |
| Baseline | 4845 | 5592 | 4661 | 931 | **90.22%** |
| DynCNN | 4845 | 5540 | 4642 | 898 | **90.01%** |

## 4.2 ON AVSS 2007

AVSS2007 is a data set for evaluating the algorithm on event detection and tracking. In this dataset, we choose six videos, which contains public surveillance scene for train station and road located in the UK, as our test data including 35,000 images. Table 3 presents the result of the performance on objects person and car with the threshold of frame differencing setting at 20. It is clear to see that the proposed dynamic convolution can preserve the precision with respect to baseline where difference is only 0.16% and 0.42% on person and car respectively.

Table 3: Average precision results on AVSS2007

| Class-Person | TotalGT | TotalPred | TruePositives | FalsePositives | **AvgPrecision** |
|---|---|---|---|---|---|
| Baseline | 227232 | 226448 | 198876 | 27538 | **80.57%** |
| DynCNN | 227232 | 222102 | 191999 | 30067 | **79.89%** |
| Class-Car | TotalGT | TotalPred | TruePositives | FalsePositives | **AvgPrecision** |
| Baseline | 44275 | 44682 | 40485 | 4195 | **89.39%** |
| DynCNN | 44275 | 44116 | 39984 | 4130 | **89.23%** |

## 4.3 ON VIRAT

VIRAT dataset collects broad surveillance videos in terms of various realism scenes including ground camera videos and aerial videos. In this dataset, 10 different scene videos on parking lot, lane and plaza totally involving 63,947 photos are selected as test data. Table 4 presents the result of the performance on objects person and car with the threshold of frame differencing setting at 20. It is clear to see that the proposed dynamic convolution can preserve the precision with respect to baseline, where the difference is only 0.54% and 0.1% on person and car respectively.

Table 4: Average precision results on VIRAT

| Class-Person | TotalGT | TotalPred | TruePositives | FalsePositives | **AvgPrecision** |
|---|---|---|---|---|---|
| Baseline | 48514 | 42154 | 36895 | 5110 | **70.50%** |
| DynCNN | 48514 | 39716 | 34733 | 4843 | **69.96%** |
| Class-Car | TotalGT | TotalPred | TruePositives | FalsePositives | **AvgPrecision** |
| Baseline | 192489 | 189550 | 173042 | 16491 | **80.94%** |
| DynCNN | 192489 | 186340 | 169027 | 17313 | **80.84%** |

## 4.4 ON YOUTUBE

The collected surveillance videos on Youtube contain 9 scene videos with different viewing angles on the zoo, MRT station, campus and street involving 115,652 images. Table 5 presents the result of the performance on objects person and car with the threshold of frame differencing setting at 20. It is clear to see that the proposed dynamic convolution can preserve the precision with respect to baseline, where the difference is only 0.7% and 0.7% on person and car respectively.

Table 5: Average precision results on Youtube

| Class-Person | TotalGT | TotalPred | TruePositives | FalsePositives | **AvgPrecision** |
|---|---|---|---|---|---|
| Baseline | 465470 | 458898 | 383821 | 75077 | **78.90%** |
| DynCNN | 465470 | 449537 | 372898 | 76639 | **78.20%** |
| Class-Car | TotalGT | TotalPred | TruePositives | FalsePositives | **AvgPrecision** |
| Baseline | 59867 | 61832 | 47447 | 14385 | **70.92%** |
| DynCNN | 59867 | 61304 | 46402 | 14905 | **70.22%** |

## 4.5 CONTINUOUS CONVOLUTION

Table 6 shows the average calculation process in each layer over all test videos of PETS 2009 dataset including the DynCNN with and without continuous convolution. The fifth and eighth column represents the new array size of dyn-convolution and the horizontal lines represent the pooling process which is used to delimit the convolution groups. As the layer more deeper, the pruned ratio generally decreases due to the increasing size of needed pixels. Note that the increasing behavior of pruned ratio in each convolution group under DynCNN (w/). In continuous convolution, the inner difference map of the first layer of convolution group directly predicts the impacted pixels of the last layer in advance, which results in that the needed pixels on the new array is more than DynCNN (w/o) and decreases gradually as layer deeper. This phenomenon can also be found in other datasets as shown in Appendix 7.4. Table 7 summarizes the average calculation amount and processing time per frame on each dataset. From this table, although DynCNN (w/o) can prune more calculation amount than DynCNN (w/), DynCNN (w/o) needs to cost more processing time on data transfer. The improvement of the overall processing time is not significant and even invalid in dataset AVSS2007 with respect to baseline. Therefore, the strategy of continuous convolution is demonstrated that can make DynCNN more valuable at the practical application as mentioned in Subsection 3.4.

Table 6: Average FLOPs result on PETS2009

| Layer-type | Maps | Baseline | | DynCNN (w/o) | | | DynCNN (w/) | | |
|---|---|---|---|---|---|---|---|---|---|
| | | wxh | FLOPs | wxh | FLOPs | %Pruned | wxh | FLOPs | %Pruned |
| Conv1-1 | 64 | 512 x 512 | 4.5E+08 | 1456x16 | 4.0E+07 | 91.1% | 1704x18 | 5.3E+07 | 88.2% |
| Conv1-2 | 64 | 512 x 512 | 9.6E+09 | 1572x16 | 9.2E+08 | 90.4% | 1702x16 | 1.0E+09 | 89.4% |
| Conv2-1 | 128 | 256 x 256 | 4.8E+09 | 978x8 | 5.9E+08 | 87.7% | 1085x10 | 8.0E+08 | 83.2% |
| Conv2-2 | 128 | 256 x 256 | 9.6E+09 | 1044x8 | 1.23E+09 | 87.2% | 1083x8 | 1.3E+09 | 86.5% |
| Conv3-1 | 256 | 128 x 128 | 4.8E+09 | 389x8 | 9.1E+08 | 81.0% | 536x12 | 1.9E+09 | 59.8% |
| Conv3-2 | 256 | 128 x 128 | 9.6E+09 | 421x8 | 1.97E+09 | 79.5% | 534x10 | 3.2E+09 | 66.5% |
| Conv3-3 | 256 | 128 x 128 | 9.6E+09 | 453x8 | 2.13E+09 | 77.8% | 532x8 | 2.5E+09 | 73.3% |
| Conv4-1 | 512 | 64 x 64 | 4.8E+09 | 181x8 | 1.69E+09 | 64.8% | 264x12 | 3.7E+09 | 22.3% |
| Conv4-2 | 512 | 64 x 64 | 9.6E+09 | 196x8 | 3.66E+09 | 61.9% | 262x10 | 6.1E+09 | 35.6% |
| Conv4-3 | 512 | 64 x 64 | 9.6E+09 | 212x8 | 3.96E+09 | 58.8% | 260x8 | 4.9E+09 | 48.9% |
| Conv5-1 | 512 | 32 x 32 | 2.4E+09 | 84x8 | 1.54E+09 | 35.8% | 118x12 | 3.3E+09 | -39.1% |
| Conv5-2 | 512 | 32 x 32 | 2.4E+09 | 89x8 | 1.64E+09 | 31.7% | 116x10 | 2.7E+09 | -13.7% |
| Conv5-3 | 512 | 32 x 32 | 2.4E+09 | 95x8 | 1.75E+09 | 27.1% | 114x8 | 2.1E+09 | 10.4% |
| FC6 FC7 | 1024 | 1 | 1.3E+08 | 1024 | 1.3E+08 | 0% | 1024 | 1.3E+08 | 0% |
| Other | | | 1.07E+10 | | 1.07E+10 | 0% | | 1.07E+10 | 0% |
| **Total** | | | **9.05e+10** | | **3.29e+10** | **63.7%** | | **4.47e+10** | **50.5%** |

Table 7: Overview results on all datasets

| Dataset | Baseline | | DynCNN (w/o) | | | DynCNN (w/) | | |
|---|---|---|---|---|---|---|---|---|
| | FLOPs | Time(ms) | FLOPs | %Pruned | Time(ms) | FLOPs | %Pruned | Time(ms) |
| PETS2009 | 9.05E+10 | 30.7 | 3.29E+10 | 63.7% | 28.3 | 4.47E+10 | 50.5% | 19.2 |
| AVSS2007 | 9.05E+10 | 30.7 | 3.91E+10 | 56.8% | 32.1 | 5.36E+10 | 40.8% | 22.9 |
| VIRAT | 9.05E+10 | 30.7 | 1.5E+10 | 83.3% | 20.3 | 2.2E+10 | 75.7% | 13.9 |
| Youtube | 9.05E+10 | 30.7 | 2.46E+10 | 72.8% | 27.5 | 4.47E+10 | 50.6% | 20 |

# 5 DISCUSSION

## 5.1 THRESHOLD OF FRAME DIFFERENCING

The threshold of IDM is used to filter the speckle noise in video. In addition, it can greatly speed up the processing time due to the high scene similarity of video—most difference values derived from frame differencing are introduced by noise. For various surveillance videos, the threshold of IDM should be different, resulting in a single threshold is difficult to be applicable to all surveillance images. To find the best threshold, we conduct a research on the relationship between the precision and calculation amount over different thresholds. Figure 4 shows the test result on 4 datasets.

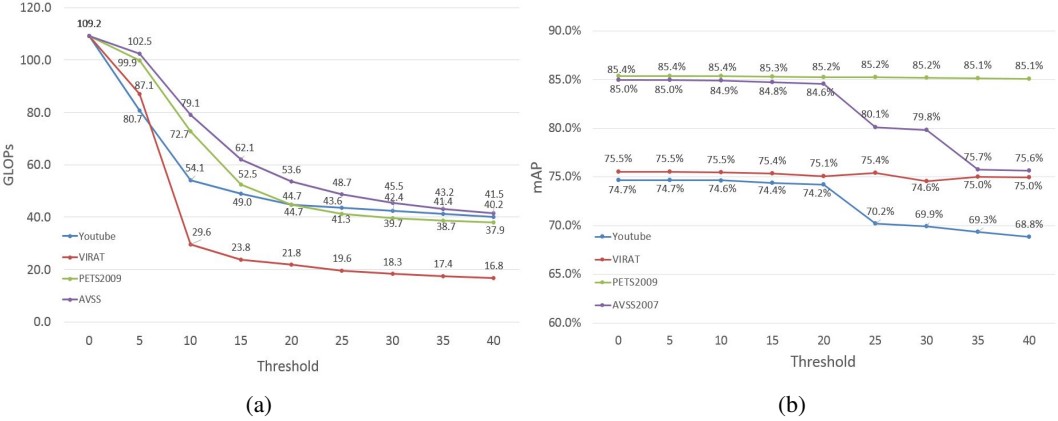

Figure 4: The average precision and calculation amount on 4 dataset over different threshold of input difference map

It is observed from Fig. 4a that the calculation amount exists a significant drop in the threshold interval of 5 to 15 over 4 datasets, which indicates the threshold greater or equal to 15 can filter the speckle noise efficiently in four datasets. However, the precision will be affected as the threshold is greater or equal to 25 as shown in Fig. 4b. Therefore, by trading the calculation amount off against precision, 20 is the best choice of threshold applicable to various surveillance videos and is selected as the default in experiments.

## 5.2 THE EFFECT ON SCENE SIMILARITY

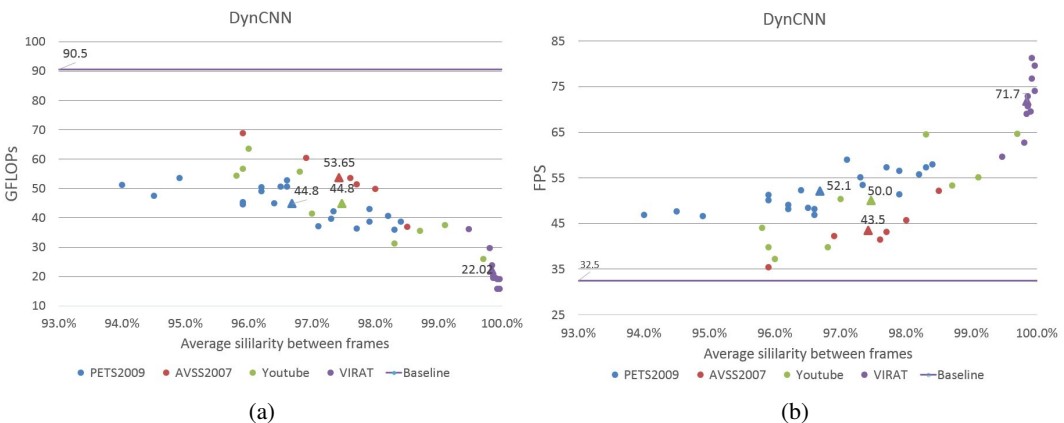

Figure 5: The distribution of each test surveillance videos

In Fig. 5, we reveal the distribution of each test surveillance videos on calculation amount and FPS with respect to the average scene similarity. The triangle mark denotes the average of videos in the same dataset while the purple line represents the value of baseline. It is obvious that the higher the

average scene similarity the more the calulation amount decreases. In addition, the FPS also tends to increase as the average scene similarity is higher, especially in the dataset VIRAT. In dataset VIRAT, the most scene of surveillance videos is parking lot where most of time are the still scene, resulting in 99% average scene similarity. The high scene similarity prunes the calculation amount of baseline model up to 75.7% (90.5 GFLOPs to 22.0 GFLOPs) and improves 2.2 times (32.5 to 71.7) of FPS. From the distribution map shown in Fig. 5, it is implied that the surveillance videos generally exist high scene similarity. Furthermore, the proposed dynamic convolution is applicable to all the videos. Therefore, it is believed that dynamic convolution can be applied to the practical applications on surveillance videos.

## 6    CONCLUSION

This paper exploits the high scene similarity generally existing in surveillance videos and proposes a dynamic convolution to reduce the calculation amount efficiently for heavy-computational surveillance analyses. The dynamic convolution only updates the feature value for the position which is predicted as changed to achieve the reuse of previous feature map. Besides the calculation amount, dynamic convolution also considers the importance of the processing time on pratical applications and employs two strategies to optimize the processing time by sacrificing a few calculation amounts. Compared with the existing technology, the it can directly be applied to the existing convolution neural network architecture without retraining and analyzing weights. Experimental results on four famous and authoritative datasets of surveillance videos provides a powerful evident that the proposed method is applicable to most of surveillance videos. The reduction on the calculation amount can be up to 75.7% while the precision preserves within 0.7% mAP. Furthermore, the processing time can be enhanced up to 2.2 times with respect to baseline. Finally, the aspect of design in the proposed method is completely different from the existing relevant methods. In the future work, we will complement our work with the existing relevent methods to verify the possibility of further acceleration.

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

# 7 APPENDIX

## 7.1 DYNAMIC CONVOLUTION MODEL

The proposed dynamic convolution is implemented on SSD512 model by caffe framework. Figure 6 and 7 illustrate the first convolution group (gp1) of SSD512 model and that with dynamic convolution (DynSDD512) respectively. In the implementation of dynamic convolution, the original model is combined with three additional layers: 1) Comparison Layer; 2) Position Layer; 3) Recovery Layer. Comparison layer is used to store the current frame and output input difference map (IDM) by comparing with the previous frame. After recieving the IDM, position layer will output the cell index table (CIT), for the position indication, the inner difference table (iDM) and the new array (NA) to next stage. Position layer generates the iDM by directly predicting the impacted pixels of the last layer in gp1 for the continuous convoltion as mentioned in subsection 3.4. Accordingly, the NA is generated according to the iDM through cell-based partition. Finally, recovery layer recieves the feature value from convolution and transfer those value where they locates on feature map through CIT.

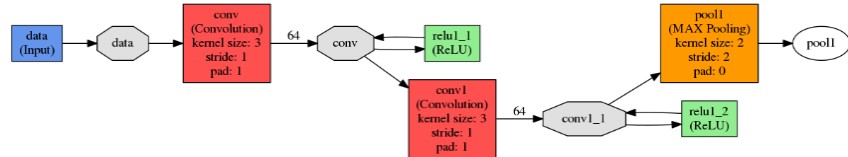

Figure 6: SSD512 model

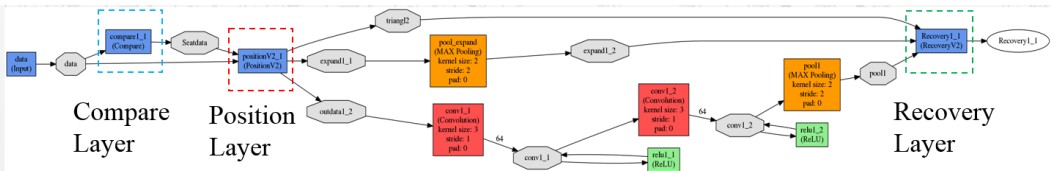

Figure 7: DynSSD512 model

## 7.2 GPU EXECUTION TIME

For a better understanding of the proposed method, we list the average execution time of each layer in detail. Take the PETS2009 dataset for example.

Table 8: Average execution time per layer

| Layer-type | Baseline (ms) | DynCNN (ms) | Saved Time% |
|---|---|---|---|
| Compare | - | 0.043 | |
| Position1 | - | 0.18 | |
| Conv1-1 | 1.15 | 0.13 | 88.2% |
| Relu1-1 | 0.58 | 0.075 | 87.0% |
| Conv1-2 | 2.53 | 0.39 | 84.5% |
| Relu1-2 | 0.58 | 0.062 | 89.4% |
| Recovery1 | - | 0.036 | |
| **Group1** | **4.84** | **0.92** | **80.9%** |
| Position2 | - | 0.27 | |
| Conv2-1 | 1.2 | 0.32 | 73.2% |
| Relu2-1 | 0.58 | 0.062 | 89.4% |
| Conv2-2 | 1.73 | 0.46 | 73.5% |
| Relu2-2 | 0.29 | 0.042 | 85.7% |
| Recovery2 | - | 0.047 | |
| **Group2** | **3.52** | **1.19** | **65.9%** |
| Position3 | - | 0.39 | |
| Conv3-1 | 0.88 | 0.60 | 31.5% |
| Relu3-1 | 0.15 | 0.059 | 60.4% |
| Conv3-2 | 1.49 | 0.76 | 49.1% |
| Relu3-2 | 0.15 | 0.048 | 67.0% |
| Conv3-3 | 1.49 | 0.42 | 71.8% |
| Relu3-3 | 0.15 | 0.04 | 72.7% |
| Recovery3 | - | 0.078 | |
| **Group3** | **4.3** | **2.4** | **44.1%** |
| Position4 | - | 0.63 | |
| Conv4-1 | 0.85 | 0.86 | -1.6% |
| Relu4-1 | 0.076 | 0.057 | 25.2% |
| Conv4-2 | 2.86 | 1.87 | 34.6% |
| Relu4-2 | 0.078 | 0.046 | 41.0% |
| Conv4-3 | 2.81 | 1.51 | 46.2% |
| Relu4-3 | 0.078 | 0.036 | 53.5% |
| Recovery4 | - | 0.16 | |
| **Group4** | **6.75** | **5.17** | **23.3%** |
| Position5 | - | 1.08 | |
| Conv5-1 | 0.78 | 1.28 | -64.5% |
| Relu5-1 | 0.014 | 0.026 | -80.9% |
| Conv5-2 | 0.77 | 0.99 | -28.4% |
| Relu5-2 | 0.014 | 0.019 | -37.3% |
| Conv5-3 | 0.77 | 0.78 | -0.9% |
| Relu5-3 | 0.014 | 0.013 | 5.6% |
| Recovery5 | - | 0.13 | |
| **Group5** | **2.36** | **4.31** | **-82.6%** |
| FC6 FC7 | - | - | 0% |
| Other | - | - | 0% |

## 7.3 EXAMPLE DETECTIONS

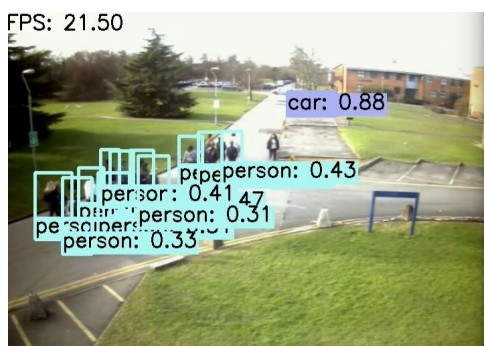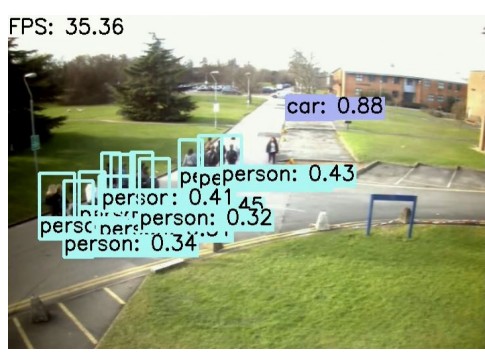

(a) Baseline on PETS2009        (b) DynCNN on PETS2009

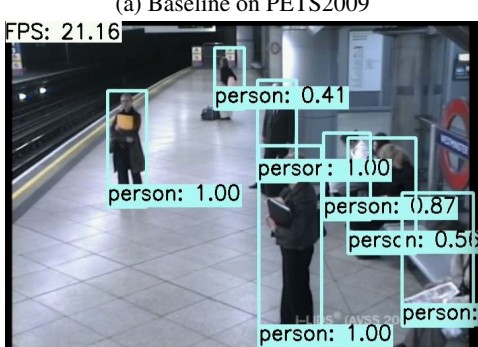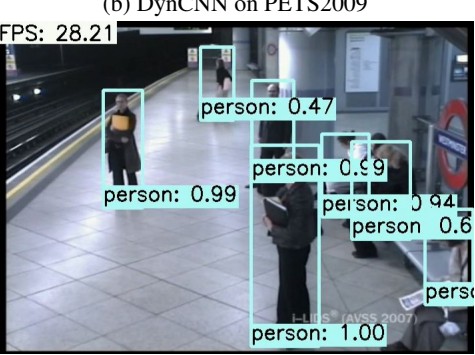

(c) Baseline on AVSS2007        (d) DynCNN on AVSS2007

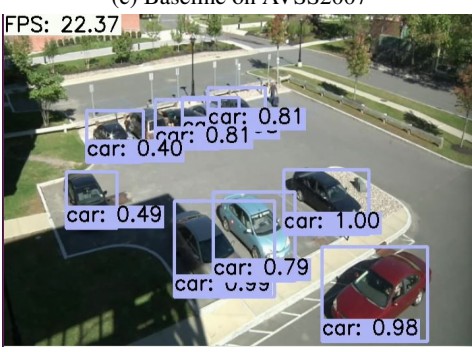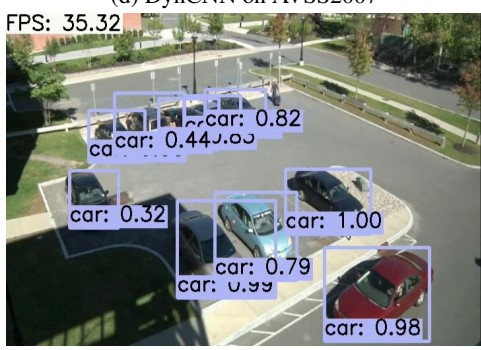

(e) Baseline on VIRAT        (f) DynCNN on VIRAT

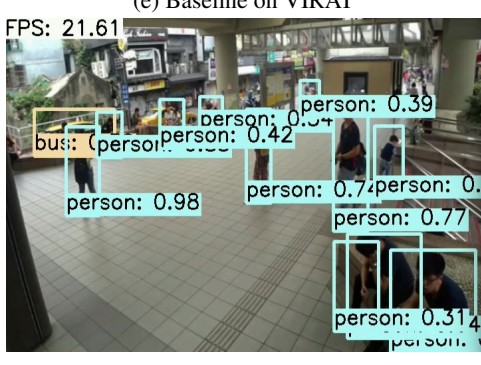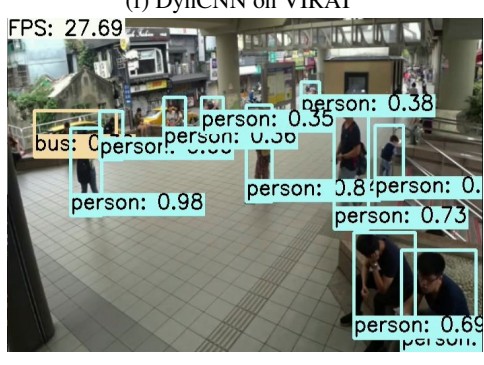

(g) Baseline on YouTube        (h) DynCNN on YouTube

Figure 8: Example detections on 4 test datasets

## 7.4 CALCULATION RESULTS ON OTHER DATASETS

Table 9: Average FLOPs result on AVSS2007

| Layer-type | Maps | Baseline | | DynCNN (w/o) | | | DynCNN (w/) | | |
|---|---|---|---|---|---|---|---|---|---|
| | | wxh | FLOPs | wxh | FLOPs | %Pruned | wxh | FLOPs | %Pruned |
| Conv1-1 | 512 x 512 | 64 | 4.5E+08 | 1704 x 16 | 4.7E+07 | 89.6% | 1931 x 18 | 6.3E+07 | 85.9% |
| Conv1-2 | 512 x 512 | 64 | 9.6E+09 | 1916 x 16 | 1.1E+09 | 88.2% | 1929 x 16 | 1.2E+09 | 87.5% |
| Conv2-1 | 256 x 256 | 128 | 4.8E+09 | 1276 x 8 | 7.5E+08 | 84.4% | 1507 x 10 | 1.1E+09 | 77.1% |
| Conv2-2 | 256 x 256 | 128 | 9.6E+09 | 1385 x 8 | 1.6E+09 | 83.0% | 1505 x 8 | 2.2E+09 | 77.0% |
| Conv3-1 | 128 x 128 | 256 | 4.8E+09 | 532 x 8 | 1.2E+09 | 74.0% | 753 x 12 | 2.6E+09 | 44.6% |
| Conv3-2 | 128 x 128 | 256 | 9.6E+09 | 578 x 8 | 2.7E+09 | 71.7% | 751 x 10 | 4.4E+09 | 53.9% |
| Conv3-3 | 128 x 128 | 256 | 9.6E+09 | 619 x 8 | 2.9E+09 | 69.7% | 749 x 8 | 3.5E+09 | 63.2% |
| Conv4-1 | 64 x 64 | 512 | 4.8E+09 | 236 x 8 | 2.2E+09 | 54.2% | 377 x 12 | 4.7E+09 | 1.7% |
| Conv4-2 | 64 x 64 | 512 | 9.6E+09 | 254 x 8 | 4.7E+09 | 50.5% | 375 x 10 | 7.8E+09 | 18.5% |
| Conv4-3 | 64 x 64 | 512 | 9.6E+09 | 273 x 8 | 5.1E+09 | 46.8% | 373 x 8 | 6.2E+09 | 35.3% |
| Conv5-1 | 32 x 32 | 512 | 2.4E+09 | 99 x 8 | 1.8E+09 | 23.8% | 129 x 12 | 3.5E+09 | -48.7% |
| Conv5-2 | 32 x 32 | 512 | 2.4E+09 | 104 x 8 | 1.8E+09 | 20.0% | 127 x 10 | 2.9E+09 | -22.1% |
| Conv5-3 | 32 x 32 | 512 | 2.4E+09 | 108 x 8 | 2.0E+09 | 16.7% | 125 x 8 | 2.3E+09 | 4.2% |
| FC6 FC7 | 1 | 1024 | 1.3E+08 | 1024 | 1.3E+08 | 0% | 1024 | 1.3E+08 | 0% |
| Other | | | 1.07E+10 | | 1.07E+10 | 0% | | 1.07E+10 | 0% |
| **Total** | | | **9.05e+10** | | **3.91e+10** | **56.8%** | | **5.36e+10** | **40.8%** |

Table 10: Average FLOPs result on VIRAT

| Layer-type | Maps | Baseline | | DynCNN (w/o)) | | | DynCNN (w/) | | |
|---|---|---|---|---|---|---|---|---|---|
| | | wxh | FLOPs | wxh | FLOPs | %Pruned | wxh | FLOPs | %Pruned |
| Conv1-1 | 512 x 512 | 64 | 4.5E+08 | 120x16 | 3.2E+06 | 99.3% | 310x18 | 9.6E+06 | 97.8% |
| Conv1-2 | 512 x 512 | 64 | 9.6E+09 | 138x16 | 8.0E+07 | 99.2% | 308x16 | 1.8E+08 | 98.1% |
| Conv2-1 | 256 x 256 | 128 | 4.8E+09 | 99x8 | 5.7E+07 | 98.8% | 188x10 | 1.4E+08 | 97.1% |
| Conv2-2 | 256 x 256 | 128 | 9.6E+09 | 110x8 | 1.3E+08 | 98.7% | 186x8 | 2.2E+08 | 97.7% |
| Conv3-1 | 128 x 128 | 256 | 4.8E+09 | 54x8 | 1.2E+08 | 97.5% | 129x12 | 4.5E+08 | 90.6% |
| Conv3-2 | 128 x 128 | 256 | 9.6E+09 | 60x8 | 2.7E+08 | 97.2% | 127x10 | 7.4E+08 | 92.3% |
| Conv3-3 | 128 x 128 | 256 | 9.6E+09 | 67x8 | 3.1E+08 | 96.8% | 125x8 | 5.8E+08 | 93.9% |
| Conv4-1 | 64 x 64 | 512 | 4.8E+09 | 36x8 | 3.2E+08 | 93.3% | 90x12 | 1.2E+09 | 74.1% |
| Conv4-2 | 64 x 64 | 512 | 9.6E+09 | 40x8 | 7.2E+08 | 92.5% | 88x10 | 2.1E+09 | 78.1% |
| Conv4-3 | 64 x 64 | 512 | 9.6E+09 | 45x8 | 8.1E+08 | 91.6% | 86x8 | 1.5E+09 | 83.7% |
| Conv5-1 | 32 x 32 | 512 | 2.4E+09 | 26x8 | 4.5E+08 | 81.1% | 60x12 | 1.6E+09 | 32.1% |
| Conv5-2 | 32 x 32 | 512 | 2.4E+09 | 28x8 | 4.9E+08 | 79.6% | 58x10 | 1.3E+09 | 45.8% |
| Conv5-3 | 32 x 32 | 512 | 2.4E+09 | 31x8 | 5.4E+08 | 77.5% | 56x8 | 1.1E+09 | 58.3% |
| FC6 FC7 | 1 | 1024 | 1.3E+08 | 1024 | 1.3E+08 | 0% | 1024 | 1.3E+08 | 0% |
| Other | | | 1.07E+10 | | 1.07E+10 | 0% | | 1.07E+10 | 0% |
| **Total** | | | **9.05e+10** | | **1.5e+10** | **83.3%** | | **2.2e+10** | **75.7%** |

Table 11: Average FLOPs result on YouTube

| Layer-type | Maps | Baseline | | DynCNN (w/o) | | | DynCNN (w/) | | |
|---|---|---|---|---|---|---|---|---|---|
| | | wxh | FLOPs | wxh | FLOPs | %Pruned | wxh | FLOPs | %Pruned |
| Conv1-1 | 512 x 512 | 64 | 4.5E+08 | 742x16 | 2.0E+07 | 95.6% | 1702x18 | 5.3E+07 | 88.2% |
| Conv1-2 | 512 x 512 | 64 | 9.6E+09 | 845x16 | 5.0E+08 | 94.8% | 1700x16 | 1.0E+09 | 89.6% |
| Conv2-1 | 256 x 256 | 128 | 4.8E+09 | 582x8 | 3.4E+08 | 92.9% | 1205x10 | 8.8E+08 | 81.7% |
| Conv2-2 | 256 x 256 | 128 | 9.6E+09 | 637x8 | 7.5E+08 | 92.2% | 1203x8 | 1.6E+09 | 82.9% |
| Conv3-1 | 128 x 128 | 256 | 4.8E+09 | 251x8 | 5.8E+08 | 87.9% | 571x12 | 2.0E+09 | 58.1% |
| Conv3-2 | 128 x 128 | 256 | 9.6E+09 | 272x8 | 1.3E+09 | 86.8% | 569x10 | 3.3E+09 | 65.2% |
| Conv3-3 | 128 x 128 | 256 | 9.6E+09 | 293x8 | 1.4E+09 | 85.7% | 567x8 | 2.6E+09 | 72.3% |
| Conv4-1 | 64 x 64 | 512 | 4.8E+09 | 113x8 | 1.0E+09 | 78.3% | 265x12 | 3.7E+09 | 22.5% |
| Conv4-2 | 64 x 64 | 512 | 9.6E+09 | 122x8 | 2.3E+09 | 76.5% | 263x10 | 6.1E+09 | 35.9% |
| Conv4-3 | 64 x 64 | 512 | 9.6E+09 | 131x8 | 2.4E+09 | 74.7% | 261x8 | 4.9E+09 | 49.1% |
| Conv5-1 | 32 x 32 | 512 | 2.4E+09 | 55x8 | 1.0E+09 | 58.3% | 110x12 | 3.0E+09 | -27.9% |
| Conv5-2 | 32 x 32 | 512 | 2.4E+09 | 59x8 | 1.1E+09 | 55.4% | 108x10 | 2.5E+09 | -4.6% |
| Conv5-3 | 32 x 32 | 512 | 2.4E+09 | 62x8 | 1.1E+09 | 52.9% | 106x8 | 1.9E+09 | 17.9% |
| FC6 FC7 | 1 | 1024 | 1.3E+08 | 1024 | 1.3E+08 | 0% | 1024 | 1.3E+08 | 0% |
| Other | | | 1.07E+10 | | 1.07E+10 | 0% | | 1.07E+10 | 0% |
| **Total** | | | **9.05e+10** | | **2.46e+10** | **72.8%** | | **4.47e+10** | **50.6%** |

