# OpenReview forum: "DynCNN: An Effective Dynamic Architecture on Convolutional Neural Network for Surveillance Videos"
_ICLR.cc/2019/Conference_

### Official Review · AnonReviewer3 · 2018-11-02
**Review for DynCNN: An Effective Dynamic Architecture on Convolutional Neural Network for Surveillance Videos**

**Rating:** 4
**Confidence:** 3

**Review:**

The paper addresses the problem of computational inefficiency in video surveillance understanding approaches. It suggests an approach called Dynamic Convolution consists of Frame differencing, Prediction, and Dyn-Convolution steps. The idea is to reuse some of the convolutional feature maps, and frame features particularly when there is a significant similarity among the frames. The paper evaluates the results on 4 public datasets. However, it just compares the approach to a baseline, which is indeed applying convnet on all frames.

- State of the art is not well-studied in the paper. Video understanding approaches usually are not just applying convnet on all frames. Many of the approaches on video analysis, select a random set of frames (or just a single frame) [5], and extract the features for them. There is another set of work on attention, that try to extracts the most important spatio-temporal [1-4] information to solve a certain task. These approaches are usually computationally less expensive than applying convnet on all video frames. I suggest the authors compare their model with these approaches.

[1] Spatially Adaptive Computation Time for Residual Networks., Figurnov et al.
[2] Recurrent Models of Visual Attention, Mnih et al.
 [3] Action recognition using visual attention, Sharma et al.
 [4] End-to-end learning of action detection from frame glimpses in videos, Yeung et al.
 [5] Two-Stream Convolutional Networks for Action Recognition in Videos, Simonyan et al.

- In addition, car and pedestrian detection performance is part of the evaluation process. In this case, the approach should be also compared to the state-of-the-art tracking approaches (that are cheaper to acquire) in terms of computational efficiency and performance.
- The writing of the paper should also improve to make the paper more understandable and easier to follow. Some examples: 1. Unnecessary information can be summarized. For example, many details on the computational costs in abstract and the introduction can just simply be replaced by stating that “these approaches are computationally costly”.  2. Using present tense for the SoTA approaches is more common.“ShuffleNet (Zhang et al. (2017)) proposed two new strategies”.  3. Long sentences are difficult to follow: “In real surveillance video application, although the calculation reduction on convolution is the main concern of speeding up the overall processing time, the data transfer is another important factor which contributes to the time”
  + The problem of large-scale video understanding is an important and interesting problem to tackle.

---

### Official Review · AnonReviewer2 · 2018-11-02
**Needs more analysis and explanation**

**Rating:** 4
**Confidence:** 4

**Review:**

Summary - This paper proposes a technique to reduce the compute cost when applying recognition models in surveillance models. The core idea is to analytically compute the pixels that changed across frames and only apply the convolution operation to those pixels. The authors term this as dynamic convolution and evaluate this method on the SSD architecture across datasets like PETS, AVSS, VIRAT.

Paper strengths
- The problem of reducing computational requirements when using CNNs for video analysis is well motivated.
- The authors analyze a standard model on benchmark datasets which makes it easier to understand and place their results in context.

Paper weaknesses
- A simple baseline that only processes a frame if \sum_{ij} D_{ij} exceeds a threshold is never mentioned or compared against. In general, the paper does not compare against any other existing work which reduces compute for video analysis, e.g., tracking. This makes it harder to appreciate the contribution or practical benefit of using this method.
- The paper has many spelling and grammar mistakes - "siliarlity", "critiria" etc.
- Continuous convolutions - It is not clear to me what is meant by this term. It is used many times and there is an entire section of results on it (Table 6), but without clearly understanding this concept, I cannot fully appreciate the results.
- Section 5.2 - what criteria or metric is used to compute scene similarity?
- Overall, I think this paper can be substantially improved in terms of providing details on the proposed approach and comparing against baselines to demonstrate that Dynamic-Convolutions are helpful.
- Design decisions such as cell-based convolution (Figure 3) are never evaluated empirically.

---

### Official Review · AnonReviewer1 · 2018-11-06
**Incremental contribution**

**Rating:** 3
**Confidence:** 4

**Review:**

In this paper, the authors propose a dynamic convolution model by exploiting the inter-scene similarity. The computation cost is reduced significantly by reusing the feature map. In general, the paper is present clearly, but the technical contribution is rather incremental. I have several concerns:
1. The authors should further clarify their advantages over the popular framework of CNN+LSTM. Actually, I did not see it.
2.  What is the difference between the proposed method and applying incremental learning on CNN?
3. The proposed method reduced the computation in which phase, training or tesing?
4. The experimental section is rather weak. The authors should make more comprehensive evaluation on the larger dataset. Currently, the authors only use some small dataset with short videos, which makes the acceleration unnecessary.

---

### Public Comment · (anonymous) · 2018-10-17
**How can I count GPU FLOPs?**

Hi,

Thanks for all your effort.
I'm trying to reproduce your work, but I can't find how to count FLOPs of each layer step. Are there any tools?
or just calculate it?

Regards.

---

> ### Author Response · Authors · 2018-10-20
> **Re: How do we count GPU FLOPs**
>
> Thanks for your comment,
>
> The GPU FLOPs calculation used in this paper focuses on the convolution layer excluding bias. The calculation of FLOPs of each layer is formulated as:
>            FLOPs = (C_i*(K^2))*H*W*C_o,
> where C_i and C_o represent the channel of input data and output feature map respectively. K^2 denotes the kernel size. H and W denote the height and width of output feature map.
>
> In our implementation, we implement the FLOPs calculation by modifying the API CuDNNConvolutionLayer<Dtype>::Forward_gpu in the file caffe/src/caffe/layers/cudnn_conv_layer.cu provided from caffe framework. Once the API finish the convolution, the FLOPs can be calculated based on the above formula. On the other hand, the layer number can be determined through the current parameter in the API such as the channels of input data and kernel.
>
> Sincerely,
> The Authors

---

### Meta-Review · Area_Chair1 · 2018-12-14
**lacking experiments against simple baselines**

**Confidence:** 5
**Recommendation:** Reject

**Metareview:**

The paper proposes a method for saving computation in surveillance videos (videos without camera motion) by re-using features from parts of the image that do not change. The results show that this significantly saves computation time, which is a big benefit, given also the amount of surveillance video input available for processing nowadays. Reviewers request comparisons to obvious baselines, e.g., selecting a subset of frames for processing or performing a low level pixel matching to select the pixels to compute new features on. Such experiments would make this paper much stronger. There is no rebuttal  and thus no ground for discussion or acceptance.